# MOTION-ALIGNED WORD EMBEDDINGS FOR TEXT-TO-MOTION GENERATION

**Ke Han**
University of Trento
ke.han.aca@gmail.com

**Yueming Lyu** *
Nanjing University
ymlv@nju.edu.cn

**Nicu Sebe**
University of Trento
niculae.sebe@unitn.it

## ABSTRACT

Existing text-to-motion (T2M) generation models typically rely on pretrained large language models to encode textual inputs. However, these models, trained on generic text corpora, lack explicit alignment between motion-related words (e.g., "clockwise", "quickly") and human skeletal movements. This misalignment, fundamentally rooted in the word embedding layers, severely limits the ability of T2M models to understand and generalize fine-grained motion semantics. To tackle this issue, we propose Motion-Aligned Text Encoding (MATE), a novel framework that explicitly incorporates motion semantics into the word embedding layers of large language models to enhance text-motion alignment for motion generation. To address the challenge of inherent semantic entanglement in motion sequences, MATE introduces two key components: 1) a motion localization strategy that establishes localized correspondences between sub-texts and motion segments, enabling soft attention guidance for semantic localization; and 2) a motion disentanglement module that isolates word-specific motion semantics via contrastive kinematic prototypes, ensuring word-level alignment between linguistic and kinematic representations. Remarkably, language models enhanced with MATE can be seamlessly integrated into existing T2M methods, significantly surpassing state-of-the-art performance on two standard benchmarks with minimal modifications. Code is available at https://github.com/ke-han-aca/MATE.git.

## 1 INTRODUCTION

Text-to-motion (T2M) generation aims to synthesize sequences of human skeletal movements conditioned on textual descriptions Zhou et al. (2024a); Chi et al. (2024); Liu et al. (2024a); Fan et al. (2024); Wang et al. (2024b). As a cross-modal generation task, T2M requires models to accurately translate textual descriptions into motion semantics and decode them into realistic human motions. However, existing T2M approaches still often exhibit limited cross-modal understanding. As shown in Fig. 1, while current models achieve the sentence-level alignment in example (1), they often struggle to robustly understand motion-related words such as "clockwise", leading to poor generalization and failure to generate plausible results for descriptions like "jogs in a clockwise motion" in example (2).

This limited word-level understanding largely stems from the limitations of text encoders, which process the textual inputs and directly determine the semantic information conveyed to the motion generator. While most T2M methods adopt pretrained large language models (LLMs) such as CLIP Radford et al. (2021) or DistilBERT Sanh et al. (2019) to leverage their strong textual understanding capabilities, these models are trained on general text corpora (or text-image pairs in the case of CLIP), lacking fine-grained alignment between motion-related words and human skeletal movements. In particular, the word embedding layers in LLMs fundamentally define the word semantics, which can differ substantially between linguistic and kinematic contexts. For instance, while "clockwise" functions linguistically as an adjective or adverb, in the kinematic domain it denotes a concrete rotational motion with a specific directional orientation. Without addressing such cross-modal word-level misalignment, LLMs struggle to encode motion-aware information effectively, inherently limiting the generation quality and generalization ability of T2M approaches.

---

*Corresponding author

(1) "a person walks in a clockwise circle"     (2) "a person jogs in a clockwise motion and falls to
                                                     their knees, he then gets back up onto his feet"

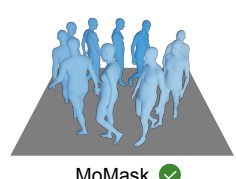
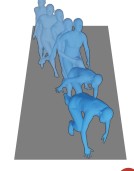
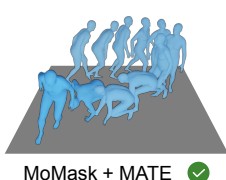

MoMask ✅              MoMask ❌              MoMask + MATE ✅

Figure 1: Examples generated by the state-of-the-art MoMask model Guo et al. (2024), with darker colors indicating motion progression. MoMask correctly produces "clockwise" motion in (1), but fails in (2), revealing limited robustness and generalization in capturing motion-related word-level semantics. Incorporating our Motion-Aligned Text Encoding (MATE) enables MoMask to produce the correct motion.

To address this limitation, we propose Motion-Aligned Text Encoding (MATE), a novel framework that incorporates motion semantics into the word embedding layers of LLMs to enhance text-motion alignment for motion generation. MATE optimizes only the word embedding layers while freezing the subsequent layers, which retain strong contextual modeling abilities acquired during large-scale language model pretraining. We hypothesize that these higher layers can generalize effectively to motion semantics, as language and motion share structural properties, i.e., both consisting of compositional elements (e.g., words and actions) organized in temporal sequences.

However, incorporating word-specific motions semantics into word embeddings remains highly challenging and largely underexplored, primarily due to the intrinsic entanglement of motion semantics. Existing datasets typically provide only sentence-level annotations for entire motion sequences Guo et al. (2022a); Plappert et al. (2016), lacking explicit alignment between specific words and corresponding motion segments. This limitation restricts the model's ability to temporally ground word-level semantics, particularly in sequences involving multiple compositional actions. More importantly, the semantics associated with related words are inherently intertwined, making it difficult to attribute distinct motion patterns to individual words, thereby limiting the model's capacity for fine-grained semantic understanding.

To address these challenges, MATE introduces two key components: 1) A *motion localization* strategy that jointly decomposes paired textual descriptions and motion sequences into semantically aligned sub-units. This enables the construction of a soft attention prior that guides the temporal localization of word semantics; and 2) A *motion disentanglement* module that isolates word-specific motion semantics through two complementary mechanisms: *self-disentanglement*, which extracts shared semantics across related motions via contrastive kinematic prototypes; and *cross-disentanglement*, which enforces the exclusion of unrelated semantics, jointly ensuring semantic purity and inter-word discriminability. The disentangled motion semantics are then aligned with their corresponding word embeddings, effectively addressing the word-level misalignment inherent in LLMs.

MATE offers an resource-efficient solution for LLM fine-tuning by optimizing only word embedding layers, while maintaining broad compatibility with various LLMs. The MATE-enhanced LLMs can be seamlessly integrated into existing T2M methods with minimal architectural modifications. Extensive experiments demonstrate that MATE consistently improves text-motion alignment and generalization capability, significantly advancing the state of the art on standard benchmarks including HumanML3D Mahmood et al. (2019) and KIT Plappert et al. (2016). The main contributions of this work are summarized as follows:

1) To the best of our knowledge, MATE is the first framework to explicitly address the text-motion misalignment fundamentally rooted in the word embeddings of LLMs for motion generation. 2) We introduce a text-motion joint segmentation strategy that automatically establishes correspondences between sub-texts and motion segments, enabling action-level semantic localization for paired text-motion data. 3) We propose a motion disentanglement module that achieves word-level semantic disentanglement, mitigating the challenge of semantic entanglement in motion sequences. 4) Extensive experiments demonstrate that MATE-enhanced language models can be seamlessly integrated into existing T2M pipelines, yielding substantial performance improvements and significantly surpassing state-of-the-art results across two standard benchmarks.

## 2 RELATED WORKS

**Text-to-Motion Generation** typically consists of two stages: text encoding and motion synthesis Shafir et al. (2024); Xie et al. (2024); Liu et al. (2023); Liang et al. (2024). In this pipeline, textual descriptions are first transformed into text embeddings, which are then mapped to motion sequences. Most existing works keep the text encoder frozen and instead focus on improving the motion generation module, often through advanced architectures such as diffusion models Tevet et al. (2023); Zhang et al. (2023b); Jin et al. (2024a); Ren et al. (2023); Wang et al. (2023) and quantized variational autoencoders Chen et al. (2023); Zhang et al. (2023a); Van Den Oord et al. (2017); Dai et al. (2024). However, these methods typically rely on off-the-shelf embeddings from pretrained large language models (LLMs), leaving unaddressed the intrinsic semantic gap between linguistic representations and kinematic motion features that arises from the LLMs themselves.

To address text-motion misalignment, some works pretrain new language models using text-motion pairs to obtain motion-aware text embeddings, such as LaMP Li et al. (2025), MotionGPT Jiang et al. (2023) and MotionGPT-2 Wang et al. (2024a). Our method differs from them in two key aspects. 1) These methods primarily operate at the sentence level, whereas our approach focuses on word-level alignment without requiring any word-level annotations, enabling modeling of finer-grained motion semantics. 2) They train new language encoders largely from scratch on relatively small text-motion datasets, which may limit their language modeling capacity. In contrast, our method fine-tunes only the lightweight word embedding layer of a pretrained LLM while keeping all higher layers frozen, thereby preserving the rich linguistic knowledge learned from large-scale corpora.

**Contrastive Learning** has been widely applied for learning text-motion aligned representations in both retrieval and generation tasks. Methods such as MotionCLIP Tevet et al. (2022), TMR Petrovich et al. (2023), TM2T Guo et al. (2022b), TEMOS Petrovich et al. (2022), and HumanTOMATO Lu et al. (2024) construct sentence-level contrastive objectives to obtain motion-aware text embeddings or to regularize motion representations. LAVIMO Yin et al. (2024) further incorporates videos as an additional modality and learns trimodal alignment through contrastive learning across text, motion, and video. More recently, LLM-based models like M$^3$GPT Luo et al. (2024) also adopt contrastive objectives during multi-task pretraining to enhance motion-language understanding.

However, these methods primarily focus on sentence-level alignment. In contrast, our method introduces a prototype-based contrastive mechanism that explicitly operates at the word level, enabling finer-grained semantic disentanglement and alignment without requiring word-level annotations.

**Large Language Model Fine-Tuning** has emerged as a powerful paradigm for adapting general-purpose textual representations to downstream tasks such as domain adaptation Ding et al. (2023); Susnjak et al. (2025), image generation Li et al. (2024); Liu et al. (2024b), and video generation Rasheed et al. (2023); Wu et al. (2023). In the context of T2M, recent works LMM Zhang et al. (2024a), AvatarGPT Zhou et al. (2024b), Motion-Agent Wu et al. (2025) unify multiple text-motion tasks within a single framework, often incorporating expanded token vocabularies and instruction tuning techniques Ouyang et al. (2022). Our method differs in two key ways: 1) Instead of fine-tuning full LLMs, which is computationally expensive, we adapt only the lightweight word embedding layers, providing a more resource-efficient approach. 2) Rather than introducing a new standalone framework, we produce plug-and-play motion-aligned text encoders that can be directly integrated into existing T2M models, improving performance without modifying their architectures.

## 3 METHOD

### 3.1 OVERVIEW

Word embedding layers, typically placed at the input of LLMs, function as a lexical lookup table that maps discrete word tokens to continuous vectors for contextual modeling. They play a crucial role in encoding word meanings and inter-word relationships. However, word-level semantics in linguistic domain often differ fundamentally from those in the kinematic domain. For example, "clockwise" and "anti-clockwise" are linguistically similar, due to sharing morphological structure and grammatical function. In contrast, they kinematically denote opposite directions of rotation and are incompatible, thus leading to significant cross-modality discrepancies.

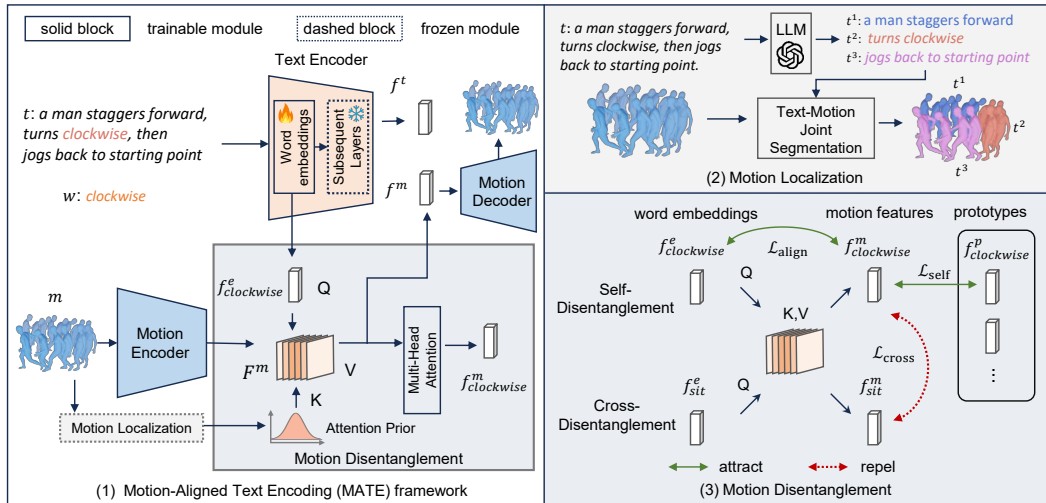

Figure 2: (1) Overview of the MATE framework, which comprises a text encoder (with trainable word embedding layers), a motion encoder, and a motion decoder, together with: (2) Motion localization, which establishes temporal text-motion correspondences to construct a Gaussian-shaped attention prior for guiding word-level semantic localization; and (3) Motion disentanglement, which employs multi-head attention to disentangle motion semantics of specific words for semantic alignment.

To bridge this gap, unlike existing T2M methods that use pretrained LLMs without adaptation, this work introduces Motion-Aligned Text Encoding (MATE), a novel approach for explicitly aligning the word embedding layers in LLMs with word-specific motion semantics.

Specifically, we formulate the learning objective as follows: given a triplet $\{t, m, w\}$ sampled from the training set, where $t$ denotes a textual description, $m$ is the corresponding motion sequence, and $w$ is a word token sampled from $t$, the goal is to align the textual semantics of $w$ with its associated motion semantics expressed in $m$ by optimizing its word embeddings in language models.

To achieve this alignment, as shown in Fig. 2, MATE comprises two key components: 1) Motion localization, which establishes correspondences between temporally aligned subtexts and motion segments for word semantic localization; 2) Motion disentanglement, which disentangles motion features that are semantically attributable to individual words for alignment with their word embeddings.

## 3.2 TEXT-GUIDED MOTION LOCALIZATION

A motion sequence is a complex integration of multiple word-level semantics, making fine-grained semantic alignment particularly challenging, especially in long sentences involving multiple words and actions. To tackle this issue, we adopt a coarse-to-fine semantic extraction strategy, with the first sub-goal of localizing the motion semantics corresponding to an individual word $w$ within the motion sequence $m$. However, most existing motion datasets Guo et al. (2022a); Plappert et al. (2016) provide only sentence-level annotations for entire sequences, lacking explicit supervision for word-level localization. To overcome this limitation, we propose a text-motion joint segmentation pipeline that automatically establishes correspondences between each sub-action described in the text and its counterpart segment in the motion sequence, as illustrated in Fig. 2 (2).

Specifically, we employ ChatGPT Roumeliotis & Tselikas (2023) to decompose each textual description $t$ into a set of sub-texts $t^1, \cdots, t^N$ ($N \geq 1$), where each sub-text describes one or more temporally coherent actions. We then seek to segment the motion sequence $m$ into $N$ non-overlapping clips, each aligned with its corresponding sub-text. To this end, we formulate an optimal partitioning problem, where segment boundaries are adjusted to minimize the matching loss between each sub-text and its corresponding motion segment. Given a sentence decomposed into $N$ sub-texts, the objective is defined as:

$$\min_{\{s_n, e_n\}} \sum_{n=1}^{N} 1 - \cos\left(\mathcal{E}_t(t^n), \mathcal{E}_m(m[s_n : e_n])\right), \tag{1}$$

where $s_n$ and $e_n$ denote the start and end frames of the $n$-th segment, constrained by $s_{n+1} = e_n$. Here, $\mathcal{E}_t$ and $\mathcal{E}_m$ are frozen text and motion encoders from a pretrained text-to-motion retrieval model Lu et al. (2024), and $\cos(\cdot, \cdot)$ denotes the cosine similarity between encoded text and motion features. An exhaustive search over all valid partitions is performed to identify the boundaries that best align motion segments with their respective sub-texts.

The obtained segmentation is not directly used as ground-truth localization, but instead serves as a soft prior to guide the discovery of the semantics of the word $\boldsymbol{w}$ within the motion sequence $\boldsymbol{m}$. Specifically, a motion encoder simultaneously extracts a sequence-level representation $\boldsymbol{f}^m \in \mathbb{R}^D$ and frame-level features $\boldsymbol{F}^m \in \mathbb{R}^{T \times D}$ from the motion sequence $\boldsymbol{m}$, where $T$ denotes the number of frames and $D$ is the feature dimension. We introduce a multi-head attention mechanism, in which the word embeddings serve as a query to explicitly attend to relevant motion features, formulated as:

$$\boldsymbol{f}^m_{\text{word}} = \text{MultiHead}(Q, K, V), \tag{2}$$

$$Q = \text{Proj}(\text{WE}(\boldsymbol{w})), \quad K = (1 + \lambda \cdot \text{AttentionPrior}(t)) \odot \boldsymbol{F}^m, \quad V = \boldsymbol{F}^m. \tag{3}$$

Here, $\text{WE}(\boldsymbol{w})$ is the trainable word embeddings of $\boldsymbol{w}$ from the text encoder, $\text{Proj}$ is a linear projection layer, and $\text{AttentionPrior}(t)$ is a temporal attention prior derived from segmentation information:

$$\text{AttentionPrior}(t) = \exp(-\frac{(t - c_n)^2}{2\sigma_n^2}), \quad \text{with} \quad c_n = \frac{s_n + e_n}{2}, \quad \sigma_n = \frac{e_n - s_n}{2}. \tag{4}$$

where $t$ is the frame index, and $s_n$, $e_n$ are the start and end frames in the localized segment of $\boldsymbol{w}$, respectively. The Gaussian-shaped $\text{AttentionPrior}(t)$ softly highlights frames near the center of the localized segment while smoothly attenuating distant frames, thereby improving robustness against localization errors introduced by the segmentation process.

### 3.3 WORD-GUIDED MOTION DISENTANGLEMENT

While the above approach enables localization of word-level semantics, the semantics of related words remain highly intertwined. For instance, as shown in Fig. 2 (1) and (2), although the segment corresponding to "turn clockwise" can be identified, it remains challenging for the model to accurately distinguish between the semantics of "turn" and "clockwise", thereby hindering the precise understanding of individual words. To address this limitation, we propose a word-guided motion disentanglement approach that explicitly isolates motion semantics attributable to individual word units. Toward this goal, we introduce the following three criteria for effective motion disentanglement.

1) **Stability**: A given word query should consistently attend to *shared* motion features across different motions that exhibit the corresponding semantics (e.g., the features disentangled by the word "clockwise" from motions of "turn clockwise" and "jog in a clockwise circle" should remain similar).

2) **Discriminability**: Different word queries should result in semantically *distinct* motion features (e.g., the features disentangled by "turn" should be distinguishable from those by "clockwise").

3) **Rationality**: Disentanglement should yield meaningful features only when the motion sequence *contains* semantics associated with the queried word (e.g., querying "clockwise" from the motion of "walk forward" should not produce a meaningful feature representation).

To satisfy Criteria 1 and 2, we introduce a self-disentanglement mechanism based on prototype representations. Specifically, we predefine a set of motion-word prototypes consisting of $K$ learnable vectors $\{\boldsymbol{f}^p_{w_k}\}_{k=1}^K$, each representing the motion semantics associated with a specific word $\boldsymbol{w}_k$. Suppose $\{\boldsymbol{t}_i, \boldsymbol{m}_i, \boldsymbol{w}_i\}$ the $i$-th triplet sample in a mini-batch, the disentangled motion features $\boldsymbol{f}^{m_i}_{w_i}$ satisfy the self-disentanglement loss $\mathcal{L}_{\text{self}}$:

$$\mathcal{L}_{\text{self}} = \frac{1}{|\mathcal{V}|} \sum_{i \in \mathcal{V}} -\log \frac{\exp(\cos(\boldsymbol{f}^{m_i}_{w_i}, \boldsymbol{f}^p_{w_i})/\tau)}{\sum_{k=1}^K \exp(\cos(\boldsymbol{f}^{m_i}_{w_i}, \boldsymbol{f}^p_{w_k})/\tau)}, \tag{5}$$

where $\mathcal{V}$ is the set of samples in a mini-batch. The loss $\mathcal{L}_{\text{self}}$ encourages $\boldsymbol{f}^{m_i}_{w_i}$ to be pulled closer to its corresponding prototype $\boldsymbol{f}^p_{w_i}$ while being pushed away from all other prototypes. Simultaneously, each prototype is optimized towards a stable and shared semantic representation across motions that express the semantics of $\boldsymbol{w}_i$. In contrast to conventional contrastive losses Radford et al. (2021); Oord et al. (2018) that operate at the batch level, the introduction of prototypes enables contrastive

learning over dataset-wide word semantics, thereby enhancing the stability and discriminability of motion disentanglement.

To satisfy Criterion 3, we further formulate a cross-disentanglement mechanism: the motion $\boldsymbol{m}_i$ is also queried by the word token $\boldsymbol{w}_j$ from the $j$-th sample, and if $\boldsymbol{m}_i$ does not contain the semantics of $\boldsymbol{w}_j$, the model is encouraged to produce motion features that are orthogonal to the reasonably disentangled features $\boldsymbol{f}_{w_i}^{m_i}$. The cross-disentanglement loss $\mathcal{L}_{\text{cross}}$ is defined as

$$\mathcal{L}_{\text{cross}} = \frac{1}{|\mathcal{N}|} \sum_{(i,j)\in\mathcal{N}} \big| \cos(\boldsymbol{f}_{w_i}^{m_i}, \boldsymbol{f}_{w_j}^{m_i}) \big| + \big| \cos(\boldsymbol{f}_{w_i}^{m_i}, \boldsymbol{f}_{w_i}^{m_j}) \big|,$$

where $\mathcal{N}$ is the set of negative pairs, and $\boldsymbol{f}_{w_j}^{m_i}$ denotes the disentangled motion features from the $i$-th sequence $\boldsymbol{m}_i$ with the word query $\boldsymbol{w}_j$ from the $j$-th sample. $\mathcal{L}_{\text{cross}}$ drives the model to discriminate whether the queried word semantics is expressed in the motion sequence to ensure the accuracy of motion disentanglement.

### 3.4 MOTION-ALIGNED WORD EMBEDDING

To align the disentangled motion features with corresponding word embeddings, we formulate an alignment loss $\mathcal{L}_{\text{align}}$ as

$$\mathcal{L}_{\text{align}} = \frac{1}{|2\mathcal{V}|} \sum_{i\in\mathcal{V}} (- \log \frac{\exp(\cos(\boldsymbol{f}_{w_i}^e, \boldsymbol{f}_{w_i}^{m_i})/\tau)}{\sum_{j\in\mathcal{V}} \exp(\cos(\boldsymbol{f}_{w_i}^e, \boldsymbol{f}_{w_j}^{m_j})/\tau)} - \log \frac{\exp(\cos(\boldsymbol{f}_{w_i}^{m_i}, \boldsymbol{f}_{w_i}^e)/\tau)}{\sum_{j\in\mathcal{V}} \exp(\cos(\boldsymbol{f}_{w_i}^{m_i}, \boldsymbol{f}_{w_j}^e)/\tau)}), \quad (6)$$

where $\boldsymbol{f}_{w_i}^e = \text{Proj}(\text{WE}(\boldsymbol{w}_i))$ is the projected word embeddings, consistent with the query in Eq (3). This loss adopts a symmetric InfoNCE formulation Oord et al. (2018) to encourage alignment between paired word embeddings and their motion semantics while simultaneously promoting separation between mismatched pairs. The motion-aligned word embedding loss is summarized as

$$\mathcal{L}_{\text{word}} = \mathcal{L}_{\text{self}} + \mathcal{L}_{\text{cross}} + \mathcal{L}_{\text{align}}. \quad (7)$$

However, the above approach primarily focuses on individual word-level alignment and overlooks the contextual dependencies among words. To tackle this issue, we further introduce a sentence-level alignment objective, which aligns the text feature vector $\boldsymbol{f}^t$ extracted from $\boldsymbol{t}$ with the corresponding motion feature vector $\boldsymbol{f}^m$ from $\boldsymbol{m}$ using an InfoNCE loss $\mathcal{L}_{\text{sent}}$. Additionally, $\boldsymbol{f}^m$ is passed through a motion decoder to reconstruct the original motion sequence $\boldsymbol{m}$, guided by a reconstruction loss $\mathcal{L}_{\text{rec}}$ to preserve detailed motion information. The overall training objective is:

$$\mathcal{L}_{\text{all}} = \mathcal{L}_{\text{rec}} + \lambda_1 \cdot \mathcal{L}_{\text{word}} + \lambda_2 \cdot \mathcal{L}_{\text{sent}}, \quad (8)$$

where $\lambda_1$ and $\lambda_2$ are weighting factors.

## 4 EXPERIMENTS

### 4.1 EXPERIMENT SETTINGS

**Dataset.** We conduct experiments on two standard motion datasets: **HumanML3D** Mahmood et al. (2019) and **KIT** Plappert et al. (2016). HumanML3D contains 14,616 motion sequences, annotated with 44,970 textual descriptions, while KIT includes 3,911 sequences paired with 6,278 descriptions.

**Evaluation Protocols.** Following standard protocols Guo et al. (2022a), we use 5 evaluation metrics: **R-Precision** and **Multimodal Distance (MMDist)** measure how accurately generated motions match the texts. **Frechet Inception Distance (FID)** evaluates the distributional similarity between generated and real motion features. **Diversity** computes the average Euclidean distance across 300 randomly sampled pairs of generated motions. **MultiModality (MModality)** reflects the variation of generated motions, calculated as the average distance among 10 motions generated from the same text.

**Implementation Details.** The text encoder in the MATE framework could be various large language models, such as CLIP Radford et al. (2021), DistilBERT Sanh et al. (2019), etc. The number of word prototypes ($K$) is automatically decided by the number of words included in the training set.

Table 1: Results on HumanML3D. "↑", "↓" and "→" indicate that higher values, lower values, or values closer to real motion are better, respectively. Red and blue highlight the top two results.

| Methods | R-Precision ↑ | | | FID ↓ | MM-Dist ↓ | Diversity → | MModality ↑ |
|---|---|---|---|---|---|---|---|
| | Top-1 | Top-2 | Top-3 | | | | |
| Real motions | $0.511^{\pm0.003}$ | $0.703^{\pm0.003}$ | $0.797^{\pm0.003}$ | $0.002^{\pm0.000}$ | $2.974^{\pm0.008}$ | $9.503^{\pm0.065}$ | - |
| GraphMotion Jin et al. (2024b) | $0.504^{\pm0.003}$ | $0.699^{\pm0.002}$ | $0.785^{\pm0.002}$ | $0.116^{\pm0.004}$ | $3.070^{\pm0.008}$ | $9.692^{\pm0.067}$ | $2.766^{\pm0.096}$ |
| Motion Mamba Zhang et al. (2024c) | $0.502^{\pm0.002}$ | $0.693^{\pm0.002}$ | $0.792^{\pm0.002}$ | $0.281^{\pm0.009}$ | $3.060^{\pm0.058}$ | $9.871^{\pm0.084}$ | $2.294^{\pm0.058}$ |
| ParCo Zou et al. (2024) | $0.515^{\pm0.003}$ | $0.706^{\pm0.002}$ | $0.801^{\pm0.002}$ | $0.109^{\pm0.005}$ | $2.927^{\pm0.008}$ | $9.576^{\pm0.088}$ | $1.382^{\pm0.060}$ |
| CoMo Huang et al. (2024) | $0.502^{\pm0.002}$ | $0.692^{\pm0.007}$ | $0.790^{\pm0.002}$ | $0.262^{\pm0.004}$ | $3.032^{\pm0.015}$ | $9.936^{\pm0.066}$ | $1.013^{\pm0.046}$ |
| BAMM Pinyoanuntapong et al. (2024a) | $0.522^{\pm0.003}$ | $0.715^{\pm0.003}$ | $0.808^{\pm0.003}$ | $0.055^{\pm0.002}$ | $2.936^{\pm0.077}$ | $9.636^{\pm0.009}$ | $1.732^{\pm0.055}$ |
| MDM Tevet et al. (2023) | $0.320^{\pm0.005}$ | $0.498^{\pm0.004}$ | $0.611^{\pm0.007}$ | $0.544^{\pm0.044}$ | $5.566^{\pm0.027}$ | $9.559^{\pm0.086}$ | $2.799^{\pm0.072}$ |
| +MATE (ours) | $0.509^{\pm0.002}$ | $0.698^{\pm0.002}$ | $0.797^{\pm0.002}$ | $0.332^{\pm0.002}$ | $3.057^{\pm0.063}$ | $9.468^{\pm0.053}$ | $2.773^{\pm0.062}$ |
| MotionDiffuse Zhang et al. (2022) | $0.491^{\pm0.001}$ | $0.681^{\pm0.001}$ | $0.782^{\pm0.001}$ | $0.630^{\pm0.001}$ | $3.113^{\pm0.001}$ | $9.410^{\pm0.049}$ | $1.553^{\pm0.042}$ |
| +MATE (ours) | $0.536^{\pm0.001}$ | $0.721^{\pm0.001}$ | $0.821^{\pm0.001}$ | $0.234^{\pm0.002}$ | $2.907^{\pm0.002}$ | $9.446^{\pm0.081}$ | $1.703^{\pm0.055}$ |
| MMM Pinyoanuntapong et al. (2024b) | $0.515^{\pm0.002}$ | $0.708^{\pm0.002}$ | $0.804^{\pm0.002}$ | $0.089^{\pm0.005}$ | $2.926^{\pm0.007}$ | $9.577^{\pm0.050}$ | $1.226^{\pm0.035}$ |
| + MATE (ours) | $0.541^{\pm0.001}$ | $0.729^{\pm0.003}$ | $0.820^{\pm0.001}$ | $0.069^{\pm0.003}$ | $2.887^{\pm0.017}$ | $9.562^{\pm0.088}$ | $1.469^{\pm0.057}$ |
| MoMask Guo et al. (2024) | $0.521^{\pm0.002}$ | $0.713^{\pm0.002}$ | $0.807^{\pm0.002}$ | $0.045^{\pm0.002}$ | $2.958^{\pm0.008}$ | $9.632^{\pm0.072}$ | $1.241^{\pm0.040}$ |
| + MATE (ours) | $0.550^{\pm0.002}$ | $0.737^{\pm0.002}$ | $0.832^{\pm0.002}$ | $0.040^{\pm0.002}$ | $2.811^{\pm0.007}$ | $9.516^{\pm0.092}$ | $1.369^{\pm0.036}$ |

Table 2: Results on KIT, using the same notations as in Table 1.

| Methods | R-Precision ↑ | | | FID ↓ | MM-Dist ↓ | Diversity → | MModality ↑ |
|---|---|---|---|---|---|---|---|
| | Top-1 | Top-2 | Top-3 | | | | |
| Real motions | $0.424^{\pm0.005}$ | $0.649^{\pm0.006}$ | $0.779^{\pm0.006}$ | $0.031^{\pm0.004}$ | $2.788^{\pm0.012}$ | $11.08^{\pm0.097}$ | - |
| GraphMotion Jin et al. (2024b) | $0.429^{\pm0.007}$ | $0.648^{\pm0.006}$ | $0.769^{\pm0.006}$ | $0.313^{\pm0.013}$ | $3.076^{\pm0.022}$ | $11.12^{\pm0.135}$ | $3.627^{\pm0.113}$ |
| Motion Mamba Zhang et al. (2024c) | $0.419^{\pm0.006}$ | $0.645^{\pm0.005}$ | $0.765^{\pm0.006}$ | $0.307^{\pm0.041}$ | $3.021^{\pm0.025}$ | $11.02^{\pm0.098}$ | $1.678^{\pm0.064}$ |
| ParCo Zou et al. (2024) | $0.430^{\pm0.004}$ | $0.649^{\pm0.007}$ | $0.772^{\pm0.006}$ | $0.453^{\pm0.027}$ | $2.820^{\pm0.028}$ | $10.95^{\pm0.094}$ | $1.245^{\pm0.022}$ |
| CoMo Huang et al. (2024) | $0.422^{\pm0.009}$ | $0.638^{\pm0.007}$ | $0.765^{\pm0.011}$ | $0.332^{\pm0.009}$ | $2.873^{\pm0.021}$ | $10.95^{\pm0.196}$ | $1.249^{\pm0.008}$ |
| BAMM Pinyoanuntapong et al. (2024a) | $0.436^{\pm0.007}$ | $0.660^{\pm0.006}$ | $0.791^{\pm0.005}$ | $0.200^{\pm0.011}$ | $2.714^{\pm0.016}$ | $10.91^{\pm0.097}$ | $1.517^{\pm0.058}$ |
| MDM Tevet et al. (2023) | $0.164^{\pm0.004}$ | $0.291^{\pm0.004}$ | $0.396^{\pm0.004}$ | $0.497^{\pm0.021}$ | $9.191^{\pm0.022}$ | $10.85^{\pm0.109}$ | $1.907^{\pm0.214}$ |
| + MATE (ours) | $0.407^{\pm0.006}$ | $0.608^{\pm0.005}$ | $0.723^{\pm0.007}$ | $0.297^{\pm0.026}$ | $2.978^{\pm0.046}$ | $10.93^{\pm0.112}$ | $1.988^{\pm0.194}$ |
| MotionDiffuse Zhang et al. (2022) | $0.417^{\pm0.004}$ | $0.621^{\pm0.004}$ | $0.739^{\pm0.004}$ | $1.954^{\pm0.062}$ | $2.958^{\pm0.005}$ | $11.10^{\pm0.143}$ | $0.730^{\pm0.013}$ |
| + MATE (ours) | $0.432^{\pm0.005}$ | $0.644^{\pm0.004}$ | $0.763^{\pm0.005}$ | $0.965^{\pm0.077}$ | $2.852^{\pm0.005}$ | $11.12^{\pm0.104}$ | $1.204^{\pm0.013}$ |
| MMM Pinyoanuntapong et al. (2024b) | $0.404^{\pm0.005}$ | $0.621^{\pm0.005}$ | $0.744^{\pm0.004}$ | $0.316^{\pm0.028}$ | $2.977^{\pm0.019}$ | $10.91^{\pm0.101}$ | $1.232^{\pm0.039}$ |
| + MATE (ours) | $0.422^{\pm0.008}$ | $0.642^{\pm0.004}$ | $0.770^{\pm0.007}$ | $0.253^{\pm0.017}$ | $2.815^{\pm0.026}$ | $10.38^{\pm0.101}$ | $1.533^{\pm0.044}$ |
| MoMask Guo et al. (2024) | $0.433^{\pm0.007}$ | $0.656^{\pm0.005}$ | $0.781^{\pm0.005}$ | $0.204^{\pm0.011}$ | $2.779^{\pm0.022}$ | $10.88^{\pm0.099}$ | $1.131^{\pm0.043}$ |
| + MATE (ours) | $0.443^{\pm0.006}$ | $0.669^{\pm0.005}$ | $0.798^{\pm0.007}$ | $0.197^{\pm0.015}$ | $2.732^{\pm0.014}$ | $10.96^{\pm0.098}$ | $1.683^{\pm0.041}$ |

Specifically, we perform lemmatization on textual descriptions and summarize the vocabulary, establishing $K = 5,161$ and $1,191$ prototypes for HumanML3D and KIT, respectively. These prototypes are initialized with their corresponding word embeddings in the pretrained language models.

After training MATE, we integrate the MATE-enhanced text encoder into existing T2M models in place of their original encoders, and retrain the models from scratch. The training and inference procedures strictly follow the official implementations of the T2M models, without any modifications. To ensure statistical reliability, we perform 20 rounds of inference and report the averaged results. Unless otherwise stated, we use MoMask Guo et al. (2024) with the MATE-enhanced CLIP as the default T2M model for evaluation.

## 4.2 COMPARISON WITH STATE-OF-THE-ART METHODS

**Quantitative Comparison.** Table 1 and Table 2 present the performance of state-of-the-art (SOTA) T2M models. The "+MATE" variants of MDM Tevet et al. (2023), MotionDiffuse Zhang et al. (2022), MMM Pinyoanuntapong et al. (2024b) and MoMask Guo et al. (2024) are obtained by retraining the original models with the pretrained CLIP text encoders replaced by MATE-enhanced CLIP, where only the word embedding layers are updated. Despite this minimal modification, our approach consistently yields substantial improvements across all evaluation metrics on both benchmarks. Notably, while the performance gains on HumanML3D are significant, the improvements on KIT are relatively modest due to the smaller dataset size, which constrains the optimization of word embeddings in large language models.

**Visualization Comparison.** Fig. 3 compares motion sequences generated by different SOTA methods. MATE accurately distinguishes fine-grained semantics, such as "kick one time with the right leg" and "three times with the left leg" in the upper example, and faithfully captures key textual descriptions like "counterclockwise circle" and "yawn" in the lower example, demonstrating the superiority of our approach in fine-grained motion semantic understanding and text-motion alignment.

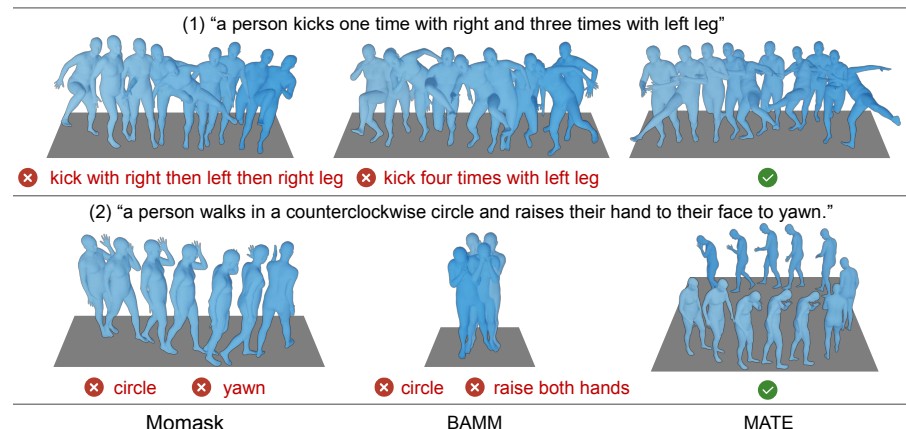

Figure 3: Comparison of motions produced by SOTA methods, with semantic misalignment in red.

Table 3: Ablation study of loss functions on HumanML3D by removing losses in Eq. (7) or (8).

| Loss Removed | Top-1 ↑ | FID ↓ | MM-Dist ↓ |
|---|---|---|---|
| $\mathcal{L}_{\text{self}}$ | 0.498 | 0.339 | 2.982 |
| $\mathcal{L}_{\text{cross}}$ | 0.533 | 0.044 | 2.934 |
| $\mathcal{L}_{\text{align}}$ | 0.519 | 0.049 | 2.954 |
| $\mathcal{L}_{\text{sent}}$ | 0.324 | 0.524 | 2.983 |
| $\mathcal{L}_{\text{rec}}$ | 0.547 | 0.042 | 2.819 |
| Full Model | **0.550** | **0.040** | **2.811** |

Table 4: Evaluations of optimizing different layers of CLIP text encoder on HumanML3D.

| Trainable Layers | Parameters | Top-1 ↑ | FID ↓ |
|---|---|---|---|
| No trainable layers | 0M | 0.521 | 0.045 |
| Word embedding layers | 3.2M | **0.550** | **0.040** |
| Subsequent layers | 37M | 0.022 | 7.611 |
| All layers | 40.2M | 0.014 | 9.468 |
| Adapter (LoRA) Zhang et al. (2024b) | 0.4M | 0.525 | 0.051 |

## 4.3 ABLATION STUDY

**Loss Functions.** Table 3 presents the ablation study of the loss functions. Removing $\mathcal{L}_{\text{self}}$, $\mathcal{L}_{\text{cross}}$, or $\mathcal{L}_{\text{align}}$ leads to varying degrees of performance degradation, highlighting their complementary and essential contributions. Specifically, $\mathcal{L}_{\text{self}}$ and $\mathcal{L}_{\text{cross}}$ promote the discriminability and effectiveness of word-level disentangled motion semantics, while $\mathcal{L}_{\text{align}}$ is critical for integrating these semantics into the word embeddings. Additionally, $\mathcal{L}_{\text{sent}}$ plays an important role by enforcing alignment between entire sentences and motion sequences, thereby modeling contextual dependencies across words. Removing $\mathcal{L}_{\text{rec}}$ also results in slight performance degradation, as motion reconstruction can enhance the informativeness of motion features.

**Optimization of Different Layers.** Table 4 compares the effects of optimizing different layers of CLIP within the MATE framework. Fine-tuning subsequent layers or the entire model markedly increases the number of trainable parameters, leading to severe overfitting and degraded generation performance due to the limited size of motion datasets. Instead, MATE restricts optimization to the word embedding layers, effectively aligning word-level semantics while preserving the strong contextual representations captured by the subsequent frozen layers. We also evaluate the LoRA, which is a commonly used LLM fine-tuning strategy Zhang et al. (2024b) by introducing additional lightweight layers while keeping the pretrained model frozen. However, it does not lead to notable performance improvements, suggesting its limited alignment ability in our setting.

Table 5: Evaluation of integrating with different large language models on HumanML3D.

| Text Encoder | Momask | | MDM | |
|---|---|---|---|---|
| | Top-1 ↑ | FID ↓ | Top-1 ↑ | FID ↓ |
| CLIP Radford et al. (2021) | 0.521 | 0.045 | 0.491 | 0.630 |
| +MATE | **0.550** | **0.040** | 0.536 | **0.234** |
| DistilBERT Sanh et al. (2019) | 0.513 | 0.053 | 0.493 | 0.615 |
| +MATE | 0.546 | 0.045 | **0.542** | 0.244 |

Table 6: Ablation of the attention prior in Eq. (4) on KIT.

| Attention Prior | Top-1 ↑ | FID ↓ | MM-Dist ↓ |
|---|---|---|---|
| No prior | 0.428 | 0.253 | 2.794 |
| Binary ($s_n \leq t \leq e_n$) | 0.431 | 0.217 | 2.746 |
| Gaussian ($s_n \leq t \leq e_n$) | 0.439 | 0.198 | 2.766 |
| Gaussian ($0 \leq t \leq T$) | **0.443** | **0.197** | **2.732** |

**Integration with Different Language Models.** We tried constructing MATE with CLIP Radford et al. (2021) and DistilBERT Sanh et al. (2019), two of the most commonly adopted language models in T2M methods, as the text encoder. As shown in Table 5, incorporating MATE with either

model consistently leads to remarkable performance gains, showing the strong compatibility and generalization capability of MATE across different language models.

**Attention Prior.** The attention prior, based on motion localization, is evaluated in Table 6. "No prior" denotes using the raw motion features as keys in Eq. (3) without any attention prior, making word-level semantic extraction from full sequences challenging. "Binary ($s_n \leq t \leq e_n$)" applies a hard binary mask, assigning 1 to frames within the target segment and 0 elsewhere, which is highly sensitive to segmentation errors. "Gaussian" is the soft attention prior defined in Eq. (4). "Gaussian ($s_n \leq t \leq e_n$)" restricts the prior within the segment, with zero attention outside. "Gaussian ($0 \leq t \leq T$)" extends the prior across the entire sequence, softly emphasizing the target region while gradually attenuating attention to neighboring frames, thus improving robustness to localization noise and achieving the best generation results.

## 4.4 VISUALIZATION RESULTS

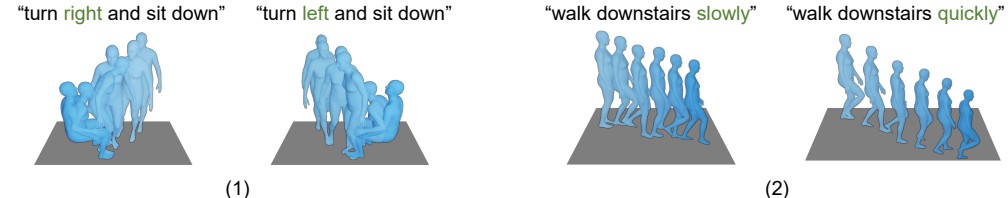

Figure 4: Motions generated by MATE when individual words in the text prompts are replaced with their antonyms. In subfigure (2), both the left and right motions span the same time period, while the right motion descends more stairs and exhibits a faster pace.

**Motion Consistency with Word Change.** To demonstrate that MATE effectively learns word-level semantic understanding, we present examples in Fig. 4, where individual words in the text prompts are replaced with their antonyms. MATE accurately captures the semantic differences between "right" and "left" as well as "slowly" and "quickly," and generates motions that are semantically aligned with the corresponding words, highlighting its robust fine-grained word-level understanding.

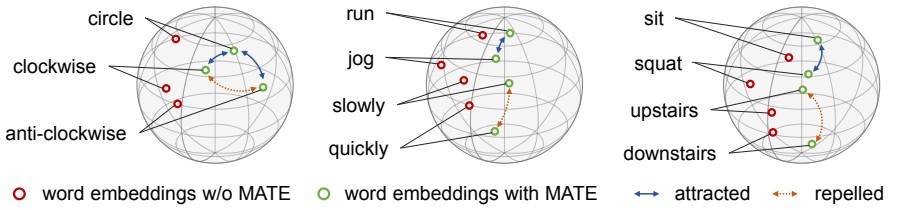

Figure 5: Word embedding distributions on the unit sphere, visualized by DOSNES Lu et al. (2019). MATE brings kinematically related words closer together while separating antonyms.

**Distributions of Word Embeddings.** To better understand the mechanism underlying the performance gains of our approach, we visualize the effect of MATE on word embeddings in Fig. 5. MATE draws together the embeddings of kinematically related motion words (e.g., "clockwise" and "circle", "run" and "jog") while pushing apart those with contrasting semantics (e.g., "quickly" and "slowly", "upstairs" and "downstairs"). This suggests that MATE structurally regularizes the word embedding space, promoting a closer alignment with motion semantics.

## 4.5 ANALYSIS OF TEXT-MOTION JOINT SEGMENTATION

Table 7: Quantitative evaluation of text-motion joint segmentation results.

| Metrics | HumanML3D | KIT |
|---|---|---|
| Text decomposition accuracy (%) | 97.0 | 98.5 |
| Motion segmentation error (sec) | 0.73 | 0.42 |

**Quantitative Evaluation.** To quantitatively evaluate text–motion joint segmentation, we manually assess the text decomposition results and annotate ground-truth segmentation boundaries for 200

motion sequences randomly sampled from the HumanML3D and KIT datasets, respectively. We then compute the average accuracy of text decomposition and the boundary errors between our predictions and the ground truth, with results reported in Table 7. The decomposition accuracy reaches 97.0% and 98.5% on the two datasets, demonstrating the strong text-processing capability of large language models within our approach. The small boundary errors (0.73s and 0.42s) further indicate that our method achieves precise alignment, with deviations well within a second, between decomposed texts and segmented motion clips. Notably, while some noise may be introduced around the segmentation boundaries, our Gaussian-based soft attention mechanism leverages soft localization signals rather than rigid hard segmentation, thereby improving robustness to such noise.

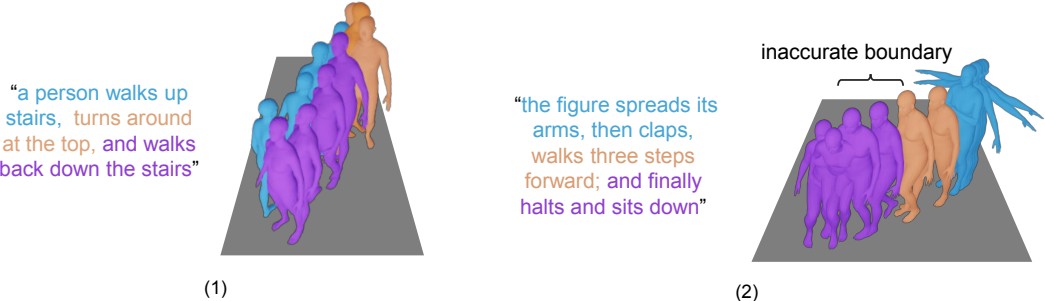

Figure 6: Examples of text-motion joint segmentation. Different colors represent the decomposed sub-texts and their corresponding motion segments. The segmentation in (1) is accurate, while in (2), the third motion segment corresponding to "halts and sits down" includes the partial semantics of the second sub-text "walks three steps", which reflects a minor boundary imprecision.

**Qualitative Analysis.** We present examples of text-motion joint segmentation in Fig. 6. The textual decomposition generated by ChatGPT is generally accurate. The primary source of error arises from imprecise boundaries predicted by the segmentation module. For instance, the boundaries in (1) are correctly identified, whereas in (2), the transition between "walks three steps forward" and "halts and sits down" is slightly misaligned. Such boundary inaccuracies are anticipated and mitigated by the Gaussian-shaped attenuation mechanism in our framework.

## 5 LIMITATIONS

1) Although our method focuses on word-level semantic alignment, certain words (e.g., "position", "starting", "area") inherently lack clear kinematic semantics or rely on contextual information. Future work will explore selective word-level semantic modeling strategies and the incorporation of contextualized queries to better handle such cases. 2) Word frequencies are imbalanced in the motion descriptions, which is not explicitly considered in this initial exploration of word embedding fine-tuning. An important direction for future work is to explore re-weighting or adaptive updating strategies to mitigate this imbalance.

## 6 CONCLUSION

In this work, we have proposed a systematic framework that integrates word-level motion localization, semantic disentanglement and alignment, addressing the text-motion misalignment fundamentally rooted in the word embeddings of LLMs for motion generation. Our approach not only demonstrates substantial improvements over state-of-the-art performance on two benchmarks, but also highlights the strong potential of word embedding fine-tuning for enabling motion-aware language modeling.

## ACKNOWLEDGMENTS

This work was supported by the EU Horizon project ELIAS (No. 101120237), the FIS project GUIDANCE (No. FIS2023-03251), the National Natural Science Foundation of China (Grant No. 62502200), the Jiangsu Provincial Science and Technology Major Project (Grant No. BG2024042), and the Natural Science Foundation of Jiangsu Province (Grant No. BK20251203).

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
