# OpenReview forum: "Motion-Aligned Word Embeddings for Text-to-Motion Generation"
_ICLR.cc/2026/Conference — ICLR 2026 Poster_

### Official Review · Reviewer_wVXx · 2025-10-31

**Soundness:** 3
**Presentation:** 3
**Contribution:** 3
**Rating:** 6
**Confidence:** 4

**Summary:**

This paper introduces Motion-Aligned Text Encoding (MATE), which improves text-to-motion generation by fine-tuning only the token-embedding layer of a frozen language model to encode motion semantics. The framework has two parts: (1) a text–motion joint segmentation that temporally localizes word meaning within motion sequences, and (2) word-guided motion disentanglement based on dataset-wide kinematic prototypes with self- and cross-disentanglement losses. Plugging the MATE-enhanced encoder into existing T2M backbones yields consistent gains on HumanML3D and KIT. Ablation studies attribute improvements to each loss component and show that restricting updates to embeddings mitigates overfitting compared with tuning deeper layers or using LoRA adapters.

**Strengths:**

1. The plug-and-play strategy of injecting motion semantics into word embeddings is simple, effective, and likely useful for future T2M and broader multimodal systems.
2. The module delivers consistent improvements across multiple T2M backbones and on both HumanML3D and KIT, demonstrating robustness and generalizability.
3. Comprehensive ablations clearly isolate the effect of each component and provide strong evidence for the overall design.

**Weaknesses:**

1. The prototype-based contrastive alignment improves stability and semantic discrimination, but may trade off representational consistency against contextual adaptability. The reported generalization to unseen words seems to rely on strong contextual cues and semantic proximity to known terms.
2. The evaluation centers on CLIP/DistilBERT encoders and does not assess more popular decoder-only language models. Testing MATE with T5-style architectures (e.g., MotionGPT [1], M3-GPT [2]) or LLaMA-based models (e.g., SoLaMi [3]) would provide a stronger case for paradigm-agnostic generalization.

>[1] Jiang, Biao, et al. "Motiongpt: Human motion as a foreign language." *Advances in Neural Information Processing Systems* 36 (2023): 20067-20079.

>[2] Luo, Mingshuang, et al. "M $^ 3$ GPT: An Advanced Multimodal, Multitask Framework for Motion Comprehension and Generation." *arXiv preprint arXiv:2405.16273* (2024).

>[3] Jiang, Jianping, et al. "Solami: Social vision-language-action modeling for immersive interaction with 3d autonomous characters." *Proceedings of the Computer Vision and Pattern Recognition Conference*. 2025.

**Questions:**

1. Relative to training the T2M backbone, what fraction of wall-clock time and compute does training the motion-aligned embeddings add?
2. The authors decompose each description with ChatGPT, which can introduce noise. Why not use datasets with human action and sentence-level annotations (e.g., BABEL [4])?

>[4] Punnakkal, Abhinanda R., et al. "BABEL: Bodies, action and behavior with english labels." *Proceedings of the IEEE/CVF conference on computer vision and pattern recognition*. 2021.

---

> ### Author Response · Authors · 2025-11-23
>
> Thank you very much for your thoughtful comments on our paper.
>
> #Response to Questions (“Q” refers to Question):
>
> _**Q1: Relative to training the T2M backbone, what fraction of wall-clock time and compute does training the motion-aligned embeddings add?**_
>
> We report the training cost comparison on a single NVIDIA RTX 4090 below:
>
> |                | Training MATE | Training MoMask with MATE |
> |----------------|---------------|----------------------------|
> | **Time**      | 1.2 h         | 17.6 h                     |
>
> Training MATE adds only 1.2/17.6 ≈ 6.8% of the wall-clock time required for training the T2M backbone. In compute terms (GPU-hours), MATE contributes less than 7% of the backbone’s total training cost. This demonstrates that MATE introduces only a small training overhead and is highly computationally efficient.
>
> _**Q2: The authors decompose each description with ChatGPT, which can introduce noise. Why not use datasets with human action and sentence-level annotations (e.g., BABEL [4])?**_
>
> We choose not to use datasets like BABEL for two main reasons.
>
> **1) Significant domain gap.**
> The textual labels in BABEL consist of short action descriptors, whereas HumanML3D and KIT contain rich, natural-language sentences describing full motion sequences. Training on BABEL would therefore result in an MATE model that does not match the vocabulary, linguistic style, or semantic granularity of the standard T2M benchmarks, leading to misalignment when applied to HumanML3D and KIT datasets.
>
> **2) Expensive annotation cost.**
> Although BABEL provides manual action-level segmentation and corresponding textual annotations, producing such fine-grained annotations is substantially more costly and time-consuming than generating the sequence-level descriptions in HumanML3D and KIT. Our method avoids this dependency by learning word-level semantic correspondences in an unsupervised manner, without requiring manual segmentation of text or motion.
>
> #Response to Weaknesses (“W” refers to Weakness):
>
> _**W1.1: The prototype-based contrastive alignment improves stability and semantic discrimination, but may trade off representational consistency against contextual adaptability.**_
>
> To address this concern, we would like to clarify that MATE explicitly preserves contextual adaptability. As described in Lines 294–297, we introduce a sentence-level alignment loss in addition to the prototype-based word alignment. This sentence-level objective preserves the ability of word embeddings to reflect sentence-level meaning, ensuring that their representations remain **context-dependent** rather than being overly constrained by a single prototype.
>
> _**W1.2: The reported generalization to unseen words seems to rely on strong contextual cues and semantic proximity to known terms.**_
>
> As shown in Fig. 8, MATE can generate more plausible motions for unseen words even **without** contextual cues, and it produces substantially more accurate motions than other models when the same contextual information is provided. This demonstrates that enhancing word-level semantic representations improves the model’s ability to generalize to unseen words and to effectively leverage contextual information, rather than relying on context alone.
>
> _**W2: The evaluation centers on CLIP/DistilBERT encoders and does not assess more popular decoder-only language models. Testing MATE with T5-style architectures (e.g., MotionGPT [1], M3-GPT [2]) or LLaMA-based models (e.g., SoLaMi [3]) would provide a stronger case for paradigm-agnostic generalization.**_
>
> We appreciate the reviewer’s suggestion. Decoder-only language models are indeed promising directions. However, these models are typically designed for multi-task motion-language learning and follow training pipelines that differ substantially from the standard T2M setting. For standard T2M models, MATE can be trained independently to fine-tune the text encoder before training the T2M backbone. In contrast, large models such as MotionGPT, M3-GPT, or SoLaMi would require incorporating MATE into their multi-stage pretraining pipelines, involving architectural and loss-function modifications. This makes a direct comparison non-trivial.
>
> Conceptually, MATE is architecture-agnostic: the word-level alignment operates on token embeddings and can be applied to any model with a learnable embedding table, including decoder-only architectures. Our work focuses on the most commonly adopted encoders in T2M pipelines (CLIP and DistilBERT), but extending MATE to generative LLM-based T2M models is an exciting direction for future research.

---

### Official Review · Reviewer_t8pa · 2025-11-01

**Soundness:** 3
**Presentation:** 2
**Contribution:** 2
**Rating:** 6
**Confidence:** 4

**Summary:**

This paper proposes MATE, a novel framework that explicitly incorporates motion semantics into the word embedding layers of large language models to enhance text-motion alignment for motion generation.

**Strengths:**

1.The paper proposes a novel framework (MATE) that explicitly addresses text-motion misalignment rooted in LLM word embeddings, a critical limitation in prior T2M methods.

2.The paper’s structure is logical, with clear problem formulation, method description, and experimental analysis.

3.Several experimental results show the effectiveness of this method.

**Weaknesses:**

1. The issue of misalignment between textual semantics and motion semantics has been raised in several works, such as LaMP. However, the authors neither compare their method with these works in the baseline nor discuss the differences between their solution and others.

2. There are numerous motion annotation errors in HumanML3D and KIT-ML datasets. For example, the motion shows the left hand being raised, while the textual annotation states the right hand. Such noisy data will have a significant impact on the method proposed by the authors.

3. The baseline lacks comparisons with LLM-based T2M methods, such as MotionGPT and MotionGPT2. The authors also fail to discuss the differences between their work and these methods.

4. The overall writing is slightly rough, and the framework diagram is not clear enough.

**Questions:**

1. Do the authors have any solutions for data with annotation errors? Such data may have many negative impacts on the training results.

2. In the demo, I am curious why the generated motion makes a larger circle when "quickly" appears in the prompt. Is this because the semantics of some actions and words have not been completely disentangled?

3.Given the failure of the re-weighting strategy for word frequency imbalance, could the authors elaborate on the differences between T2M word embedding alignment and standard long-tail learning? Are there adaptive update strategies (e.g., dynamic weighting based on prototype stability) that could mitigate this issue?

4.Could the authors compare MATE with recent LLM-based T2M methods (e.g., MotionGPT2, MotionGPT) in terms of model complexity, training cost, and fine-grained semantic alignment ability? This would better highlight MATE’s advantages.

5.Author should discuss with other works on the misalignment between motion and language question.

I hope author can increase the presentation quality and answer my questions.

---

> ### Author Response · Authors · 2025-11-23
> **Response to Questions**
>
> We are grateful for your valuable comments on this paper.
>
> #Response to Questions (“Q” refers to Question):
>
> _**Q1: Do the authors have any solutions for data with annotation errors?**_
>
> We adopt a prototype-based semantic learning framework that inherently reduces the impact of annotation errors. Because each word prototype is learned from numerous text-motion pairs, the influence of mislabeled samples is greatly diluted, making annotation errors have only a limited effect on the overall training process.
>
> _**Q2: In the demo, I am curious why the generated motion makes a larger circle when "quickly" appears in the prompt.**_
>
> We find that the variation in circle size is largely due to randomness introduced by different sampling seeds, rather than the model systematically associating “quickly” with a larger circle. Across seeds, the generated motions consistently show the intended semantics of “quickly”.
>
> _**Q3.1: Could the authors elaborate on the differences between T2M word embedding alignment and standard long-tail learning?**_
>
> In long-tail classification, the objective is to improve recognition accuracy for rare classes by amplifying their gradients. In contrast, our task aims to align word embeddings with motion semantics while preserving the rich linguistic priors already encoded in the pretrained language model. This leads to two key differences:
>
> **1) The target of learning is a semantic transformation, not a class decision boundary.**
> High-frequency words in our setting carry crucial or diverse motion contexts and thus form stable prototypes. Reducing their update weights (as done in classical long-tail methods) disrupts the learning of generalized motion semantics rather than correcting imbalance.
>
> **2) Low-frequency words should remain consistent with pretrained language priors.**
> Unlike long-tail classes that benefit from stronger updates, low-frequency words in our setting rely heavily on the pretrained embedding space for semantic stability. Over-amplifying their gradients easily causes semantic drift, undermining both the linguistic meaning and motion alignment.
>
> _**Q3.2: Are there adaptive update strategies (e.g., dynamic weighting based on prototype stability) that could mitigate this issue?**_
>
> Existing adaptive re-weighting techniques are designed for long-tail classification. As far as we know, they typically rely on sample frequency rather than prototype stability, and are therefore not directly suitable for our setting, where the goal is semantic alignment rather than class re-balancing. The idea of using prototype properties, such as prototype stability or alignment confidence, to modulate update strength is indeed promising, and we regard it as a valuable direction for future work.
>
> _**Q4: Could the authors compare MATE with recent LLM-based T2M methods (e.g., MotionGPT2, MotionGPT) in terms of model complexity, training cost, and fine-grained semantic alignment ability?**_
>
> The comparison on HumanML3D is shown below, and also included in Appendix B.2 in the updated paper (already uploaded).
>
> | Models        | Trainable Parameters | Training Cost       | Top 1 ↑ | FID ↓  |
> |---------------|-----------------------|----------------------|---------|--------|
> | MotionGPT     | 220M                  | 8 × Tesla V100       | 0.492   | 0.232  |
> | MotionGPT2    | 89M                   | 4 × NVIDIA A100      | 0.496   | 0.191  |
> | MATE (Ours) | 3.2M             | 1 × NVIDIA 4090   | 0.550 | 0.040 |
>
> MATE achieves substantially better generation performance while requiring far fewer trainable parameters and significantly lower training cost. Unlike MotionGPT and MotionGPT-2, which rely on large-scale pretraining for multi-task integration, our approach offers a lightweight, efficient, and effective framework for text-motion alignment and generation.
>
> _**Q5: Author should discuss with other works on the misalignment between motion and language.**_
>
> Thank you for the suggestion. We have added a discussion of the relevant works in the Related Work section (Lines 115–123) in the updated paper, and we also include the added text below for your reference.
>
> _To address text-motion misalignment, some works pretrain new language models using text-motion pairs to obtain motion-aware text embeddings, such as LaMP, MotionGPT, and MotionGPT-2. Our method differs from them in two key aspects. 1) These methods primarily operate at the sentence level, whereas our approach focuses on word-level alignment without requiring any word-level annotations, enabling modeling of finer-grained motion semantics. 2) They train new language encoders largely from scratch on relatively small text-motion datasets, which may limit their language modeling capacity. In contrast, our method fine-tunes only the lightweight word embedding layer of a pretrained LLM while keeping all higher layers frozen, thereby preserving the rich linguistic knowledge learned from large-scale corpora._

---

> ### Author Response · Authors · 2025-11-23
> **Response to Weaknesses**
>
> #Response to Weaknesses (“W” refers to Weakness):
>
> _**W1: The issue of misalignment between textual semantics and motion semantics has been raised in several works, such as LaMP. However, the authors neither compare their method with these works in the baseline nor discuss the differences between their solution and others.**_
>
> Thank you for pointing out this issue. We have added a discussion of prior works on text-motion alignment in the updated paper. Please also refer to our Response to Q5.
>
> In addition, we provide a direct comparison with LaMP in the table below. “H3D→KIT” indicates that the text encoder is pretrained on HumanML3D and then the model is trained on KIT, which evaluates the text encoder’s language modeling capacity and cross-dataset generalization ability.
>
> | Methods | Top 1 ↑ (H3D) | FID ↓ (H3D) | Top 1 ↑ (H3D→KIT)  | FID ↓ (H3D→KIT) |
> |---------|---------|--------|----------------------|--------------------|
> | LaMP    | 0.557   | 0.032  | 0.423                | 0.226              |
> | MATE    | 0.550   | 0.040  | 0.441                | 0.203              |
>
>
> MATE and LaMP achieve comparable results on HumanML3D. However, MATE exhibits stronger generalization in the H3D→KIT setting. We attribute this to the design difference:
> 1) MATE fine-tunes only the lightweight word embedding layer of LLMs while keeping all higher layers frozen, thereby preserving the rich linguistic knowledge learned from large-scale corpora.
> 2) LaMP, in contrast, learns its text encoding solely from limited text-motion data, which restricts its language modeling capacity and weakens its generalization to new datasets.
>
> _**W2: There are numerous motion annotation errors. Such noisy data will have a significant impact.**_
>
> Please refer to our Response to Q1 for a detailed explanation.
>
> _**W3: The baseline lacks comparisons with LLM-based T2M methods, such as MotionGPT and MotionGPT2. The authors also fail to discuss the differences between their work and these methods.**_
>
> Please refer to Response to Q4 for a detailed comparison with LLM-based T2M methods, and to Response to Q5 for a discussion of the key differences between our approach and these works.
>
> _**W4: The overall writing is slightly rough, and the framework diagram is not clear enough.**_
>
> Thank you for the suggestion. We have improved both the writing and the framework diagram for better clarity in the updated paper (already uploaded).

---

### Official Review · Reviewer_XYL8 · 2025-11-01

**Soundness:** 2
**Presentation:** 3
**Contribution:** 2
**Rating:** 4
**Confidence:** 5

**Summary:**

This paper addresses a fundamental challenge in text-to-motion (T2M) generation. Existing approaches that rely on pretrained large language models (LLMs) often suffer from a semantic misalignment between motion-related vocabulary—such as “clockwise” or “quickly”—and the corresponding human skeletal movements. This issue stems from the generic nature of text embeddings in LLMs, which are not inherently designed to capture fine-grained kinematic semantics. As a result, the ability of T2M models to accurately interpret and generalize nuanced motion concepts remains limited.

To tackle this problem, the authors propose a novel framework named Motion-Aligned Text Encoding (MATE), which aims to explicitly inject motion semantics into the word embedding layer of LLMs, thereby enhancing the alignment between linguistic expressions and motion representations.

The MATE framework consists of two core components:

* A motion localization strategy, which jointly decomposes paired text descriptions and motion sequences into semantically aligned sub-units. This process establishes a soft attention prior that facilitates the temporal grounding of word-level semantics in motion sequences.

* A motion disentanglement module, which isolates word-specific motion semantics through two complementary mechanisms:
    * self-disentanglement, which extracts shared semantics across related motions using contrastive kinematic prototypes;
    * cross-disentanglement, which explicitly excludes irrelevant semantics to ensure discriminability between different motion words.

The disentangled motion semantics are then aligned with their corresponding word embeddings, effectively mitigating the word-level misalignment inherent in conventional LLMs.

Extensive experiments validate the effectiveness of the MATE framework. The authors demonstrate that MATE-enhanced language models can be seamlessly integrated into existing T2M pipelines, leading to significant performance gains over state-of-the-art methods across multiple standard benchmarks.

**Strengths:**

The problem this paper focuses is critical for text-to-motion generation task. This paper is well-structured. In addition, the paper provide extensive experiments to prove the effectiveness of the proposed method. The proposed MATE is integrated into several existing T2M pipelines to validate its effectiveness.

**Weaknesses:**

## Major Concern:
My primary concern pertains to the foundational text-to-motion matching strategy outlined in Section 3.2. The proposed method relies on a pre-trained dual encoder to partition motion sequences into sub-clips by minimizing a matching loss. While this approach is intuitively reasonable, its effectiveness is critically dependent on the capacity and quality of the pre-trained encoders.

The dual encoder adopted from HumanML3D, in my view, may possess inherent limitations for fine-grained semantic alignment. These limitations stem from its relatively simple architecture, limited model scale, and—most importantly—the nature of its pre-training data.
The HumanML3D dataset provides only sentence-level motion-text pairs, lacking explicit, fine-grained alignment between sub-motion segments and their corresponding word-level descriptions. Consequently, I have significant doubts regarding the encoder's ability to accurately infer the optimal start and end boundaries for sub-motion clips based solely on weak, sentence-level supervision.
An inaccurate partitioning in this initial stage could propagate errors throughout the entire disentanglement and alignment pipeline.

This challenge is particularly acute in the domain of human motion generation, which is notoriously data-scarce, especially for fine-grained annotations. The situation stands in stark contrast to other domains like action detection or fine-grained action recognition, where similar matching strategies have succeeded. In those fields, methods typically benefit from powerful, large-scale dual encoders (e.g., CLIP) trained on massive, diverse datasets, which provide a robust semantic foundation.
A similar replication of this paradigm in the data-poor motion generation domain, without a similarly robust backbone, represents a significant and potentially under-addressed risk in the current work.

## Other Concern
* **Novelty:** As indicated in my major concern, the core technical components—including fine-grained clip recognition, text-segment pairing, and contrastive learning for alignment—have been extensively explored and validated in other research areas (e.g., video-text alignment). While the integration and application of these ideas to the T2M generation task is valuable, the conceptual framework itself does not appear to be fundamentally novel. Therefore, I am inclined to assign a more moderate score regarding methodological innovation.
* **Discussion of Related Work:** To better position this work within the broader research landscape, I would recommend the authors to include a discussion of contrastive learning-based methods from related fields.
* **Clarification of Figure 2:** Figure 2 (3) is a bit unclear.

**Questions:**

Please refer to the weakness.

---

> ### Author Response · Authors · 2025-11-23
>
> Thank you very much for your constructive comments.
>
> #Response to Weakness (“W” refers to Weakness):
>
> _**W1: I have significant doubts regarding the encoder's ability to accurately infer the optimal start and end boundaries for sub-motion clips based solely on weak, sentence-level supervision.**_
>
> Thank you for raising this concern. We clarify this issue as follows.
>
> **1) Our method does _NOT_ rely on optimal or accurate sub-motion clip boundaries.**
>
> The motion localization stage is designed only to provide **coarse semantic partitioning**, which is then converted into a **soft Gaussian attention prior** for guiding word-level semantic disentanglement. Importantly, the motion is **not** physically segmented into hard clips, and the entire sequence remains intact. The Gaussian prior gently biases attention toward relevant temporal regions rather than enforcing strict boundaries.
>
> The improvements reported in Table 6 demonstrate that this soft attention formulation is effective and robust to coarse localization boundaries.
>
> **2) Experimental evidence shows reliable alignment under sentence-level supervision.**
>
> Although the model is trained only with sentence-motion pairs, it consistently infers meaningful internal structure within each motion sequence. Both quantitative and qualitative results in Appendix D.1 show low segmentation errors. Supplementary video visualizations further confirm that the inferred boundaries are stable and semantically coherent.
>
> **3) Data scarcity does not hinder our segmentation module, because it does not require generalization to unseen data.**
>
> While the motion domain is indeed data-scarce, our segmentation module operates in a **closed-set** setting: It is trained on sentence-level text-motion pairs from HumanML3D/KIT, and then performs segmentation **on exactly the same data**. Thus, the dual encoder is only required to infer sub-motion structure within the motions it has already learned at the sentence level, rather than generalizing to unseen data. This makes the problem significantly easier than large-scale, open-domain action localization. Even without CLIP-scale pretraining, the encoder is sufficiently expressive to capture coarse motion boundaries on this closed-set distribution.
>
> In summary, our method is intentionally designed to tolerate coarse localization, avoid error propagation, and operate reliably under sentence-level supervision and limited data.
>
>
> _**W2:  Novelty: The core technical components have been extensively explored and validated in other research areas (e.g., video-text alignment).**_
>
> While prior work in other domains has explored fine-grained clip recognition and contrastive alignment, these techniques are generally developed for **sentence-level** alignment. In contrast, the central novelty of our work lies in enabling **word-level motion semantic disentanglement and alignment** for T2M generation, which has not been explored in prior works.
>
> 1) To achieve this, we introduce a soft Gaussian attention prior for motion localization, specifically designed to support **word-specific motion semantic disentanglement** without word-level annotations.
>
> 2) We propose a **prototype-based word-level semantic alignment** approach to directly optimize the word embedding layer of a pretrained LLM.
>
> Therefore, while certain technical concepts may appear related to those in other domains, the problem setting, modeling challenges, and proposed solution in our work are fundamentally distinct and constitute a novel contribution to the T2M field.
>
>
> _**W3:  Discussion of Related Work: I would recommend the authors to include a discussion of contrastive learning-based methods from related fields.**_
>
> Thank you for the suggestion. We have added a discussion in the Related Work section (Lines 124–134) in the updated paper, and we include the added text below for your reference.
>
> _Contrastive Learning has been widely applied for learning text–motion aligned representations in both retrieval and generation tasks. Methods such as MotionCLIP, TMR, TM2T, TEMOS, and HumanTOMATO construct sentence-level contrastive objectives to obtain motion-aware text embeddings or to regularize motion representations. LAVIMO further incorporates videos as an additional modality and learns trimodal alignment through contrastive learning across text, motion, and video. More recently, LLM-based models like M3-GPT also adopt contrastive objectives during multi-task pretraining to enhance motion–language understanding._
>
> _However, these methods primarily focus on sentence-level alignment. In contrast, our method introduces a prototype-based contrastive mechanism that explicitly operates at the word level, enabling finer-grained semantic disentanglement and alignment without requiring word-level annotations._
>
> _**W4: Figure 2 (3) is a bit unclear.**_
>
> Thank you for pointing this out. We have revised Figure 2 (3) to improve its clarity in the updated paper (already uploaded).

---

### Official Review · Reviewer_NpWK · 2025-11-03

**Soundness:** 4
**Presentation:** 4
**Contribution:** 3
**Rating:** 8
**Confidence:** 4

**Summary:**

This paper presents Motion-Aligned Text Encoding (MATE), a novel framework aimed at improving the semantic alignment between textual descriptions and human motion in text-to-motion (T2M) generation tasks. The authors introduce a lightweight fine-tuning strategy that updates only the word embedding layer of a pre-trained language model, while keeping all other layers frozen. This enables the embedding space to better capture motion-relevant semantics.

MATE consists of two core components:
- Motion localization, which aligns motion sub-sequences with corresponding text spans through soft attention guidance; and
- Motion disentanglement, which employs contrastive learning with kinematic prototypes and cross-word negative sampling to disentangle word-specific motion features.

Experimental results show that MATE can serve as a plug-and-play module for various T2M frameworks, significantly enhancing the performance of state-of-the-art models such as MoMask and MotionDiffuse on HumanML3D and KIT benchmarks. Additional ablation, visualization, and user studies further validate the effectiveness and generalizability of the proposed approach.

**Strengths:**

1. The paper effectively identifies a major limitation in existing T2M approaches, word-level semantic misalignment, and addresses it through a simple yet effective embedding-layer fine-tuning mechanism.
It is the first to tackle the systemic mismatch between linguistic and motion semantics from the word embedding perspective, offering both scientific motivation and engineering value.

2. By restricting fine-tuning to the word embeddings, the method remains computationally efficient and easily integrable into existing architectures.

3. The proposed motion localization and disentanglement modules enhance the interpretability of semantic learning, and the contrastive objectives are elegantly designed.

4. The experimental section is comprehensive, which These experiments strongly support the method’s efficiency.
- Evaluation on multiple T2M baselines (e.g., MoMask, MMM, MotionDiffuse) demonstrate consistent improvements on HumanML3D and KIT datasets across metrics such as R-Precision and FID.
- Visualization of embedding distributions further illustrates improved discrimination of action-related terms such as “left/right” and “quickly/slowly.”
- The ablation studies thoroughly examine loss design, parameter settings, module compatibility, attention mechanisms, and semantic disentanglement.
- Supplementary experiments—including integration with different LLMs, user evaluations, and cross-task tests (retrieval, editing, completion)—confirm the stability and versatility of MATE.

**Weaknesses:**

1. Text segmentation using ChatGPT may introduce stochasticity and hinder replication.

2. The reliance on a fixed, limited set of prototypes could constrain generalization, potentially biasing the learned representations toward dataset-specific distributions.

3. Direct fine-tuning of the text encoder might reduce its capacity to retain rich or nuanced semantics, especially for ambiguous expressions.

4. While embeddings are improved, the inference stage still does not explicitly address temporal misalignment, leaving potential issues with word ordering or motion duration.

**Questions:**

1. If ChatGPT-assisted segmentation is unstable, how does the model handle inconsistent segmentation results?
Additionally, When a motion segment corresponds to multiple keywords, how does the system ensure these cues jointly guide generation? Does it produce two motion sequences (each with its own representation $f^m$) or merge them into a unified representation?

2. Line 246 states "we predefine a set of motion-word prototypes consisting of K learnable
vectors". Does “predefine” mean that the prototypes are manually specified or that only the number K is fixed while the prototypes are learned automatically? If the former, how are the prototype words selected?

3. Was any negative transfer observed for non-motion-related words  during word embedding adjustment? Are there mechanisms to mitigate this?

4. Could MATE’s capability of cross-domain generalization be evaluated through few-shot or zero-shot tasks, i.e., testing on unseen verbs, domains, or motion styles, rather than relying solely on the original training distribution?

**Details Of Ethics Concerns:**

The paper does not involve human subjects, personal data, or sensitive information. It uses publicly available motion–language datasets (e.g., HumanML3D, KIT) and focuses on model alignment techniques. No identifiable, private, or ethically sensitive content is introduced.

---

> ### Author Response · Authors · 2025-11-23
>
> We sincerely appreciate your thorough and constructive review of our work.
>
> #Response to Questions (“Q” refers to Question):
>
> _**Q1.1: If ChatGPT segmentation is unstable, how does the model handle inconsistency?**_
>
> We observed that ChatGPT occasionally produces slightly different segmentation granularities for the same text. For example, “a person runs forward, stops and bends down” may be segmented into
>
> (1) 3 sub-texts (“runs forward”, “stops”, “bends down”), or
>
> (2) 2 sub-texts (“runs forward”, “stops and bends down”).
>
> Our segmentation module is designed to naturally adapt the number of motion clips to match sub-texts, resulting in 3 and 2 segments for (1) and (2), respectively, ensuring consistent alignment regardless of segmentation granularity.
>
> Importantly, such cases are extremely rare. Among 100 randomly sampled sentences across 5 temperature settings, only 3 showed this variation.
>
> _**Q1.2: When a motion corresponds to multiple keywords, how does the system ensure these cues jointly guide generation? Does it produce two sequences or merge them?**_
>
> Different keywords are used as **separate** guidance signals, i.e., the system produces two motion features for two keywords, each aligned with its corresponding word.
>
> _**Q2. "we predefine a set of motion-word prototypes consisting of K learnable vectors". Does “predefine” mean the prototypes are manually specified or only the number K is fixed while the prototypes are learned automatically? If the former, how are the prototype words selected?**_
>
> K is determined by the number of unique words in the training vocabulary, and the prototype set is automatically formed from these vocabulary words rather than being manually specified. The prototypes are fully learnable, and each one is initialized with the pretrained embedding of its corresponding word.
>
> _**Q3. Was any negative transfer observed for non-motion-related words? Are there mechanisms to mitigate this?**_
>
> We did not observe noticeable negative transfer on non-motion-related words. This is largely because we explicitly exclude high-frequency non-motion words (e.g., a, person, man, be) from the update process. By preventing updates to these embeddings that lack motion semantics, the model avoids unintended shifts in their representations, effectively mitigating potential negative transfer.
>
> _**Q4. Could MATE’s capability of cross-domain generalization be evaluated through few-shot or zero-shot tasks, i.e., testing on unseen verbs, domains, or motion styles?**_
>
> Evaluating cross-domain generalization quantitatively is challenging due to format incompatibilities across datasets. Nevertheless, we provide qualitative evidence in the Appendix:
> 1) User study (B.1): Human preference results indicate strong cross-domain generalization, with MATE consistently outperforming prior models.
> 2) Example analysis on unseen words (B.5): MATE produces more plausible motions for unseen words, compared with existing methods.
>
> #Response to Weaknesses (“W” refers to Weakness):
>
> _**W1: Text segmentation using ChatGPT may introduce stochasticity and hinder replication.**_
>
> We observed that the segmentation results are highly stable for two reasons.
> 1) We design detailed segmentation rules in prompts to constrain the model’s behavior and minimize randomness (Appendix C).
> 2) Text segmentation itself is a simple and well-defined linguistic preprocessing task, for which LLMs exhibit highly deterministic behavior when given clear instructions.
>
> We will release the complete segmented corpus to ensure full reproducibility.
>
> _**W2: The reliance on a fixed, limited set of prototypes could constrain generalization, potentially biasing the learned representations toward dataset-specific distributions.**_
>
> Despite a fixed prototype set, both the word embeddings and the prototypes are adapted from pretrained language embeddings toward motion semantics. This design enables MATE to inherit broad linguistic priors from the pretrained encoder while learning motion-specific refinements. As shown in our Response to Q4, MATE achieves stronger generalization than existing methods, indicating that the use of prototypes does not constrain, and in fact enhances, generalization.
>
>
> _**W3: Direct fine-tuning of the text encoder might reduce its capacity to retain rich or nuanced semantics.**_
>
> We only fine-tune the lightweight word embedding layer to enhance word-level grounding for motion semantics without altering the higher-level linguistic knowledge encoded in the deeper layers. However, we acknowledge that fully quantifying how much semantic capacity is preserved is non-trivial, and we will further investigate this in future work.
>
> _**W4: While embeddings are improved, the inference stage still does not explicitly address temporal misalignment.**_
>
> Thank you for your insightful comment. Our current work focuses on enhancing word-level embeddings, and we consider explicit temporal modeling a valuable direction for future exploration.

---

### Meta-Review · Area_Chair_2sYP · 2026-01-01

**Summary:**

This paper proposes Motion-Aligned Text Encoding (MATE): a lightweight appraoch that fine-tunes only the word embedding layer of a frozen LLM, together with (i) text–motion localization that yields a soft attention prior, and (ii) prototype-based word-guided motion disentanglement, and shows it can be plugged into existing T2M pipelines with gains on HumanML3D/KIT. It recieved mixed scores, one accept (8), two marginal accept (6), and one marginal reject (4). Across reviews, the following main concerns are raised:
1. **Reliance on segmentation / boundary quality and weak supervision.** Reviewer XYL8 questioned whether the dual-encoder localization can reliably infer clip boundaries from sentence-level supervision and worried about error propagation through the pipeline. This is also the main concern of reviewer XYL8.
2. **Novelty / positioning vs prior alignment methods.** XYL8 argued key components resemble ideas used in video-text alignment, and asked for stronger positioning and related-work discussion.
3. **Missing comparisons and clarity issues.** Reviewer t8pa asked for comparisons/discussion vs LaMP and LLM-based T2M methods (MotionGPT/MotionGPT2), plus writing/diagram clarity.
4. **Robustness / reproducibility concerns.** Reviewers NpWK, t8pa and wVXx raised concerns about (a) ChatGPT-assisted text decomposition and reproducibility and (b) the method’s sensitivity to annotation noise in benchmarks.

**Reviewer Concerns:**

### Concerns addressed by the rebuttal / revision
1. **Segmentation/boundary dependence (XYL8)**
* Authors clarified the method does not require optimal boundaries; localization provides coarse partitioning used only to form a soft Gaussian attention prior (no hard motion splitting), and they argue robustness to imperfect localization.

2. **Novelty/positioning (XYL8)**
* Authors emphasized the core contribution as word-level disentanglement/alignment for T2M (vs typical sentence-level alignment) and added related-work discussion.

3. **Missing comparisons and clarity issues.**
* Authors added (i) an explicit LaMP comparison including a cross-dataset generalization setting and (ii) a comparison to MotionGPT / MotionGPT2 including training cost/parameters.

4.**ChatGPT segmentation stability & reproducibility (NpWK/wVXx)**
* Authors provided evidence of limited variability across temperature settings and stated they will release the segmented corpus for reproducibility.

The authors also provided compute overhead (wVXx) analysis which states training MATE as ~6.8% of the bacbone training wall-clock time and <7% GPU-hours in their setting, as well as revision to increase clarity (t8pa).

### Concerns outstanding.
* **Core skepticism about whether weakly supervised localization is “reliable enough” (XYL8)** The rebuttal reframes the dependency (soft prior, closed-set segmentation), but this may not fully satisfy the reviewer’s central worry about the semantic strength of the dual encoder learned from sentence-level data.

* **External validity / broader generalization of prototypes and word-level alignment** Authors provide some qualitative evidence for unseen words/cross-domain behavior and argue prototypes do not constrain generalization, but this aspect remains less conclusively validated than the in-distribution benchmark improvements. These are not quantitatively validated.

* **Annotation noise robustness (t8pa)** The response argues prototype learning dilutes mislabeled samples, but there is no clear dedicated experiment quantifying robustness under controlled noise.

In addition, i personally feel figure 2 needs further improvement for clarity. For now, it's hard to grasp the main idea and workflow. I also think the qualitative visualizations can be polished. Right now all visualizations are in stick figure, and the temporal segmentation visualization are rather hard to browse because of the asyncronized time line.

**Reviewer Scores:**

The majority of reviewers vote for acceptance of this work. From the rebuttal, I can see that reviewer NpWK's concerns (i.e., reproducibility/prototype questions) have been clarified --- this reviewer likely remain the score (8). Reviewer t8pa would be satisfied with the added experiment and comparisons, and possibly slightly raise the score (6-> 6/7). Reviewer wVXx would retain the original score (6). Reviewer may still hold the borderline score as the reviewer did not fully address the weak dual encoder question.

Given these, I lean to recommend acceptance of this manuscript but with relatively low confidence. The authors are encouraged to further polish the visualization quality and presentation clarity (e.g., figure 2).

---

### Decision · Program_Chairs · 2026-01-26

Accept (Poster)